# Pairwise GUI Dataset Construction Between Android Phones and Tablets

**Han Hu, Haolan Zhan,**\* **Yujin Huang, Di Liu**
Monash University
Melbourne, Australia
{han.hu, haolan.zhan, yujin.huang}@monash.edu, dliu0024@student.monash.edu

## Abstract

In the current landscape of pervasive smartphones and tablets, apps frequently exist across both platforms. Although apps share most graphic user interfaces (GUIs) and functionalities across phones and tablets, developers often rebuild from scratch for tablet versions, escalating costs and squandering existing design resources. Researchers are attempting to collect data and employ deep learning in automated GUIs development to enhance developers' productivity. There are currently several publicly accessible GUI page datasets for phones, but none for pairwise GUIs between phones and tablets. This poses a significant barrier to the employment of deep learning in automated GUI development. In this paper, we introduce the Papt dataset, a pioneering pairwise GUI dataset tailored for Android phones and tablets, encompassing 10,035 phone-tablet GUI page pairs sourced from 5,593 unique app pairs. We propose novel pairwise GUI collection approaches for constructing this dataset and delineate its advantages over currently prevailing datasets in the field. Through preliminary experiments on this dataset, we analyze the present challenges of utilizing deep learning in automated GUI development.

## 1 Introduction

Mobile apps are ubiquitous in our daily life for supporting different tasks such as reading, chatting, and banking. Smartphones and tablets are the two types of portable devices with the most available apps [49]. To conquer the market, one app is often available on both smartphones and tablets [41]. Due to comparable functionalities, the smartphone and tablet versions of the same app have a highly similar Graphical User Interface (GUI). Popular apps always share a similar GUI design between phone apps and tablet apps, for example, YouTube [55] and Spotify [48]. If a tool can automatically recommend a GUI design for the appropriate tablet platform based on existing mobile GUIs, it can significantly minimise the developer's engineering effort and accelerate the development process. From the user side, a comparable design could provide data or study on user preferences for consistency across different devices. It reduces the need for them to learn new navigation and interaction patterns [44]. Therefore, automated GUI development tasks, such as the cross-platform conversion of GUI designs, GUI recommendations, etc., are gradually gaining attention from industry and academia [57, 37, 12]. However, the field of automatic GUI development is still in a research bottleneck, lacking breakthroughs and widely recognized tools or methods. If a present developer needs to develop a tablet-compatible version of their app, they usually start from scratch, resulting in needless cost increases and wasted existing design resources.

According to our observations, the growth of automated GUI development is hindered by two reasons. First, as deep learning approaches, particularly generative models [50], become more widespread in the field of automated GUI development, researchers increasingly need a pairwise, high-quality GUI

---

\*corresponding author

37th Conference on Neural Information Processing Systems (NeurIPS 2023) Track on Datasets and Benchmarks.

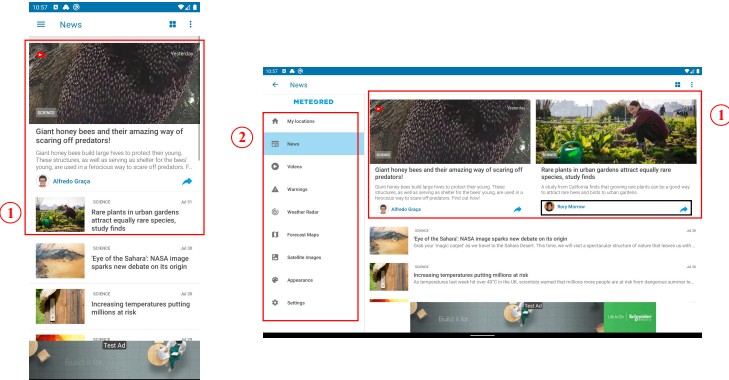

Figure 1: An example of a phone-tablet GUI pair of the app 'BBC News'. The GUI on the left is from the phone, while the GUI on the right is from the tablet.

dataset for crucial purposes such as training models, and analyzing patterns. The current datasets, for example, Rico [15], ReDraw [43], and Clay [37], only include single GUI pages with UI metadata and UI screenshots, and there are no pairwise corresponding GUI page pairs. Current datasets are only suitable for GUI component identification, GUI information summarising, and GUI completion. The lack of valid GUI pairs in current datasets has severely hindered the growth of GUI automated development.

Second, the collecting of GUI pairs between phones and tablets is more labor-intensive. Individual GUI pages can be automatically collected and labeled by current GUI testing and exploration tools [25, 42] to speed up the collection process. However, due to the disparity in screen size and GUI design between phones and tablets, it is challenging to automatically align the content on both GUI pages. Figure 1 shows an example of a phone-tablet GUI pair of the app 'BBC News' [7]. To accommodate tablet devices, the UI component group 1 in the phone GUI is converted to the GUI parts marked as 1 in the tablet GUI. We can find that these components not only change their positions and sizes but also the contents and types of GUIs. Another UI component group, which is marked as 2 in the tablet GUI, is not present in the phone GUI at all. One tablet GUI page may correspond to the contents of multiple phone GUI pages due to the different screen sizes and design styles. The UI contents in group 2 of the tablet GUI correspond to other phone GUIs.

In this paper, we introduce our novel pairwise GUI data collection methodologies and present the newly curated Papt dataset. Papt is the first **PA**irwise GUI dataset between Android **P**hones and **T**ablets, encompassing 10,035 GUI pairs from 5,593 app pairs. The initiative commences with an articulate delineation of the data collection methods, coupled with a comprehensive statistical analysis of the dataset. Thereafter, we expound upon the specific structuring of the data, highlighting the intrinsic advantages conferred by this unique organization. Progressing into the experimental phase, we undertake preliminary investigations, expressly tailored for the GUI conversion task. Finally, we engage in a discussion of the challenges currently faced by existing models in the realm of GUI conversion, shedding light on the complexities inherent to this domain.

A novel dataset, featuring pairwise GUI representations spanning both mobile and tablet platforms, presents itself as an indispensable resource in the evolving realm of automated GUI design research. Specifically, this dataset is instrumental in cross-platform and adaptive design initiatives [27, 32], serving as a foundation for model training, verification, and empirical analysis. It further delivers tangible data samples grounded in real-world contexts, accelerating advancements in user experience augmentation [31] and bolstering accessibility features [39, 57]. Moreover, it can establish a unified benchmark for stringent testing and validation in related domains [20, 29]. Beyond its tangible utility in practical developmental domains, the dataset also enriches the theoretical domain, pushing the boundaries of contemporary interface design research.

In summary, the novel contributions of this paper are the following:

- We contribute the first pairwise GUI dataset between Android phones and tablets [2].

---

[2] https://github.com/huhanGitHub/papt

- We demonstrate our pair collection approaches and open source our data collection tool with the intention to foster further research and facilitate the collection of customized GUI data by future researchers [3].
- We perform preliminary experiments on the dataset and report the experimental results. We discuss the challenges faced by the current models on the GUI conversion task.

## 2    Related Work

**Android GUI Dataset** Numerous datasets have been curated to support deep learning applications in mobile app design. Notably, Rico [15], with 72,219 screenshots from 9,772 apps, serves as the cornerstone for GUI modeling. Despite its prominence, Rico's limitations, such as noise and erroneous labeling [16, 35], have spurred further dataset enhancements. Enrico [36] is the first enhanced dataset drawn from Rico, which is used for mobile layout design categorization and consists of 1460 high-quality screenshots produced by human revision, covering 20 different GUI design topics. The VINS dataset [9], developed specifically for detecting GUI elements, comprises 2,740 Android screenshots manually collected from various sources including Rico and Google Play. Since the data cleaning process for both the Enrico and VINS datasets involves humans, adopting such approaches to improve existing Android GUI datasets at scale is expensive and time-consuming. In contrast, CLAY [37] leverages deep learning for dataset noise reduction, producing 59,555 screenshots based on Rico. Beyond Rico derivatives, several works [10, 11, 12, 13, 53, 26] also build their datasets for various GUI-related tasks such as Skeleton Generation, search and component prediction.

Yet, contemporary Android GUI datasets grapple with obsolescence due to infrequent updates. More critically, the absence of pairwise GUI pages between distinct mobile devices, such as phones and tablets, severely constrains the progress of end-to-end tasks, inclusive but not limited to GUI generation and GUI search. So, we introduce the first pairwise dataset between Android phones and tablets to fill the current gaps.

**Layout Generation** Facilitating GUI conversion between Android and tablet apps underpins our dataset's intent. We briefly overview contemporary layout generation methods. LayoutGAN [38], pioneering generative layouts, employs GANs with self-attention and introduces a differentiable wireframe rendering layer for CNN-based discrimination. LayoutVAE [33] is an autoregressive model, utilizing LSTM [24] to process UI elements and VAEs [34, 56] for layout creation. VTN [6] builds on the VAE structure, replacing encoder-decoder with Transformers [52] for layout learning sans annotations. At the same time, LayoutTransformer [22], a purely Transformer-based framework, is proposed for layout generation. It captures co-occurrences and implicit relationships among elements in layouts and uses such captured features to produce layouts with bounding boxes as units. Based on our experiment results, we find that current layout generation models have limited capacity for GUI conversion between Android and tablet layouts, suggesting the potential research direction in automated GUI development.

## 3    Methodologies of Data Collection

In the process of data collection, our methodology is bifurcated into two phases: the algorithmic automatic pairing of phone-tablet GUI pages, followed by manual verification of these pairings. In this section, we first introduce the source app pairs of our dataset in subsection 3.1. Then, we demonstrate two employed distinct techniques for matching GUI pairs: by dynamically adjusting the device resolution (as detailed in subsection 3.2), and by computing similarity metrics (elaborated upon in subsection 3.3). Subsequent to the automated pairing, pairs are rigorously scrutinized by a panel of three experts, each possessing a minimum of one year of GUI development expertise, to discard any inaccuracies or mismatches, which is described in subsection 3.4.

### 3.1    Source App Pairs

We first crawl 6,456 tablet apps from Google Play. Then we match their corresponding phone apps by their app names and app developers. Finally, we collect 5,593 valid phone-tablet app pairs from 22 app categories as the data source for this dataset. The three most common categories of apps

---

[3] https://github.com/huhanGitHub/papt/tree/main/tools

in the data source are: *Entertainment* (8.87%), *Social* (7.04%), and *Communication* (5.83%). The categories of apps in our data source exhibit a balanced and dispersed distribution. The majority of these categories account for approximately 4% to 6% of the entire dataset, reinforcing the dataset's robustness in terms of generalizability and heterogeneity. An exhaustive breakdown of app category distribution is available in the supplementary materials.

## 3.2 Dynamically Adjusting Resolution for GUI Pairing

In the Android ecosystem, adaptive layouts enable a tailored user experience that caters to a spectrum of screen dimensions, ensuring app compatibility across phones, tablets, foldable and ChromeOS devices, as well as supporting varied orientations and resizable configurations [1]. Consequently, certain apps instantiate device-specific layout files, contingent upon the resolution of the host device. Notably, such apps deploy identical APK files across both mobile and tablet platforms. Our initial focus is directed towards this category of apps.

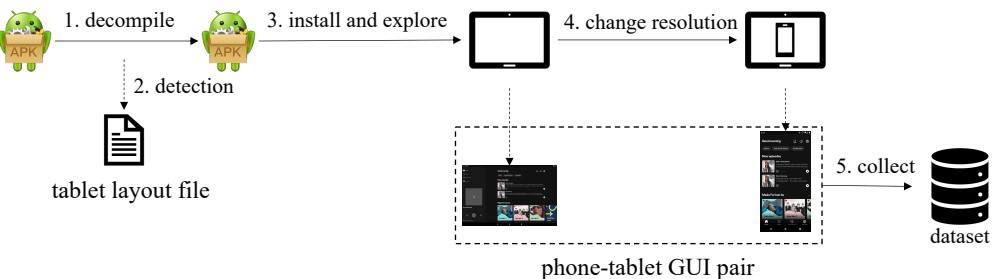

Figure 2: Pipeline of dynamically adjusting the resolution for GUI pairing

According to Android's official guidelines for supporting different screen sizes [18], Android developers can provide alternate layouts for displays with a minimum width measured in density-independent pixels (dp or dip) by using the smallest width screen size qualifiers. Illustratively, for a *MainActivity*, one might encounter layouts like *res/layout/main_activity.xml* for smartphones and *res/layout-sw600dp/main_activity.xml* for tablets exhibiting a width of 600 density-independent pixels. It is pivotal to understand that the qualifier (*sw600dp*) references the more diminutive dimension among the screen's two axes, unaffected by device orientation. Such dedicated layouts indicate that the respective app's *MainActivity* is tailored for a tablet-centric interface. Notably, two prevalent smallest width qualifiers, 600dp and 720dp, are meticulously designed to align with the prevalent 7" and 10" tablets [18].

Figure 2 shows the pipeline of matching GUI pairs by dynamically adjusting device resolutions. Drawing from our initial analyses, the initial steps involve decompiling the acquired app pairs and probing for layout files that cater to the 600dp and 720dp tablet configurations within the source files (as depicted in steps 1 and 2 of Figure 2). Out of the assessed 5,593 app pairs, layout files characteristic to the smallest width qualifiers for tablet devices are identifiable in 1,214 pairs, all of which utilized the same app across phone and tablet platforms. It becomes evident that such pairs employ Responsive/Adaptive layouts, which modulate the GUI layout contingent on the device they are hosted on. For the identified app pairs, the Windows Manager Command of ADB (*adb shell wm size*) [4] is employed to dynamically alter the device's resolution, thereby revealing their intended smartphone and tablet GUI layouts. By sequentially emulating both the tablet and phone resolutions on a single tablet device, we facilitate the automatic pairing of GUIs captured pre and post these simulations, as illustrated in steps 4 and 5 of Figure 2.

## 3.3 UI Similarity-Based GUI Pairing

Excluding 1,214 applications with platform-specific XML definitions, we apply UI similarity-based GUI pairing to other 4,379 phone-tablet pairs, accommodating those developing mirror apps or utilizing Java/Kotlin code for dynamic screen adaptation.

As depicted in Figure 3, the process for matching GUI pairs is primarily hinged on the comparison of UI similarities. Notably, since each app comprises numerous GUI pages, a direct pairwise

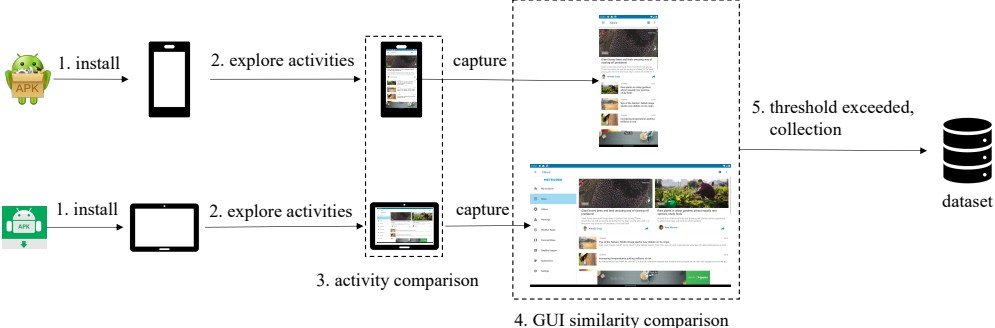

Figure 3: Pipeline of similarity-based GUI pairing

comparison of all pages across mobile and tablet apps becomes computationally intensive. In the Android ecosystem, an activity serves as the basic window for the app to render its UI [3]. Initially, corresponding Android activity pairs for each app pair are discerned. This is followed by a comparison of individual GUI pages rooted in their associated Android activity pairs. The methodology employed to ascertain Android activity pairs involves the extraction of activity names from each GUI, which are subsequently encoded into semantic vectors via a pre-trained BERT model [17]. Pairs are then matched based on the proximity of their semantic vectors, as delineated in Step 3 of Figure 3. For instance, GUIs in activities titled *homeActivity* and *mainActivity* can be correlated based on the similarity of their vectors. It's noteworthy that a single Android activity can encompass multiple fragments [2] and GUI pages [40, 21], each distinguished by its unique set of UI components and layout structures in prevalent applications. To refine the pairing process at a lower granularity, the attributes of GUI components in activity pairs are contrasted between mobile and tablet devices, a process elucidated in Step 4 of Figure 3. Herein, UI components are primarily distinguished based on their typologies and inherent properties. Components from both mobile and tablet platforms, exhibiting analogous types and attributes, are subsequently deemed as the same views. As an illustrative point, two coinciding *TextViews*, *ImageViews*, or *Buttons* with congruent texts or images are recognized as paired entities. If more than half of the UI components in two GUI pages are paired, they are considered a phone-tablet GUI pair (Step 5 in Figure 3).

### 3.4 Manual GUI Pair Verification

We collect 12,331 GUI pairs in the automated collection. Due to the inherent constraints of ADB, contemporary data collection mechanisms frequently falter in retrieving metadata for the UI type *WebView* as well as certain bespoke third-party UI components. Concurrently, for certain obscured Android fragments and UIs, only the metadata for the foreground UI is of relevance, given that the overlaid UI remains undisplayed on the prevailing screen. Yet, extant data collection tools [4, 51] might inadvertently accumulate data from both the visible and concealed UI, leading to potential data inaccuracies.

To address these challenges, an additional round of manual data validation is undertaken by a panel of three specialists, each boasting a minimum of one year's expertise in Android development. These individuals meticulously scrutinize each pair against two pivotal metrics: data integrity and pair coherence. This manual validation encompass a meticulous verification of harvested pairs, identification and elimination of pairs with erroneous metadata, and an assessment of the logical consistency of the matched pairs. Throughout this rigorous verification phase, the evaluators also actively engage in the manual matching of certain phone-tablet GUI pairs. In this round of manual verification, we remove 2,296 pairs of logical inconsistency and obvious visual mismatches between screenshots and their metadata.

## 4 Characteristics of the Papt Dataset

Using the methodologies described in Section 3, we have gathered 10,035 valid phone-tablet GUI pairs. In this section, we elucidate key features of the Papt dataset encompassing UI View types (elaborated in subsection 4.1), data format (detailed in subsection 4.2), and comparative strengths

relative to other datasets (expanded upon in subsection 4.3). Comprehensive statistics of the dataset are available in the supplementary material.

## 4.1 Distribution of UI View Types

Within Android development, UI views serve as fundamental elements for user interface construction, responsible for both rendering and handling user interactions within a designated screen segment [5]. For instance, diverse elements such as buttons, textual content, images, and lists all fall under the category of views. Notably, in our dataset, *TextView* and *ImageView* types and their derivatives, like *AppCompatTextView*, *AppCompatCheckedTextView*, and *AppCompatImageView*, emerge as the most prominent UI view categories. Their respective counts, 73,349 and 68,496, overshadow other view types. This dominance can be attributed to the GUI's inherent focus on information dissemination predominantly through text and images. Additionally, *ImageButton* and *Button*, tallying 10,366 and 10,235 respectively, rank as the third and fourth most prevalent UI views. Given the pivotal role of click operations in user-GUI interactions, the ubiquity of button-related views in our dataset is logical.

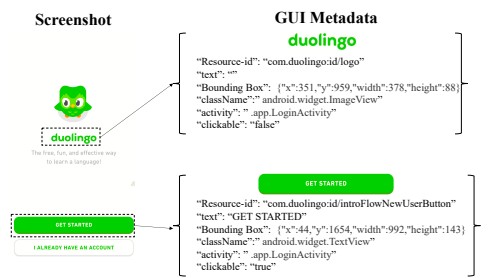

Figure 4: A screenshot of a GUI and its part UI metadata

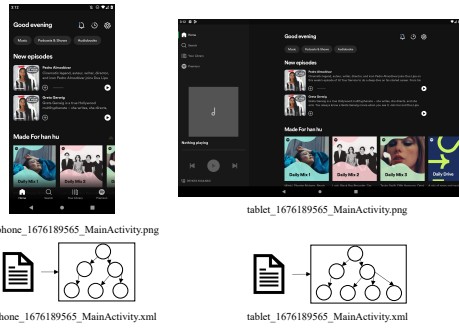

Figure 5: An example GUI pair in Spotify

## 4.2 Data Format

In this section, we first show an example screenshot of a GUI and its corresponding UI metadata. Then, we introduce the format of each GUI pair in the dataset.

**GUI Screenshot and its Metadata** We install and run phone-tablet app pairs on the Pixel6 and Samsung Galaxy tab S8, respectively. We use uiautomator2 [51] to collect screenshots and GUI metadata of the dynamically running apps. Figure 4 shows an example of a collected GUI screenshot and metadata of some UI components inside the GUI. This example is from the app 'Duolingo [19]. The metadata is a documentary object model (DOM) tree of current GUIs, which includes the hierarchy and properties (e.g., class, bounding box, layout) of UI components. We can infer the GUI hierarchy from the DOM tree hierarchy in metadata.

**GUI Pair** Figure 5 shows an example of pairwise GUI pages of the app 'Spotify' in our dataset. All GUI pairs in one phone-tablet app pair are placed in the same directory. Each pair consists of four elements: a screenshot of the GUI running on the phone (*phone_1676189565_MainActivity.png*), the metadata data corresponding to the GUI screenshot on the phone ( *phone_1676189565_MainActivity.xml* ), a screenshot of the GUI running on the tablet (*tablet_1676189565_MainActivity.png* ), and the metadata data corresponding to the GUI screenshot on the tablet (*tablet_1676189565_MainActivity.xml* ). The naming format for all files in the dataset is *Device_Timestamp_Activity Name*. As shown in Figure 5, The filename *tablet_1676189565_-MainActivity.xml* indicates that this file was obtained by the tablet and was collected with the timestamp *1676189565*, this GUI belongs to *MainActivity* and this file is a metadata file in XML format. We use timestamps and activity names to distinguish phone-tablet GUI pairs.

**Pairs in the Dataset** The pairs in the dataset are contained in separate folders according to the app. Most of the app folders are named after the package name of the app's APK, for example, *air.com.myheritage.mobile* , and a few are named after the app's name, for example, *Spotify*. In each app folder, as described in Section 4.2, each pair contains four elements: the phone GUI screenshot,

the XML file of the phone GUI metadata, the corresponding tablet GUI screenshot, and the XML file of the tablet GUI metadata. We also shared the script for loading all GUI pairs in the open source repository.

## 4.3   Comparison with Available Datasets

Compared to other current datasets, our dataset has advantages in terms of applicable tasks and data accuracy.

Table 1: Comparison between our dataset and other GUI datasets

| Dataset | GUI Platform | #GUIs | #Data Source App | Latest Updates | Mainly supported tasks |
|---|---|---|---|---|---|
| Rico [15] | Phone | 72,000 | 9,700 | Sep. 2017 | UI Component Recognition, GUI completion |
| UI2code [10] | Phone | 185,277 | 5,043 | June. 2018 | UI Skeleton Generation |
| Gallery D.C. [11] | Phone | 68,702 | 5,043 | Nov. 2019 | UI Search |
| LabelDroid [13] | Phone | 394,489 | 15,087 | May. 2020 | UI Component Prediction |
| UI5K [12] | Phone | 54,987 | 7,748 | June. 2020 | UI Search |
| Enrico [36] | Phone | 1,460 | 9,700 | Oct. 2020 | UI Layout Design Categorization |
| VINS [9] | Phone | 2,740 | 9,700 | May. 2021 | UI Search |
| Screen2Words [53] | Phone | 22,417 | 6,269 | Oct. 2021 | UI screen summarization |
| Clay [37] | Phone | 59,555 | 9,700 | May. 2022 | UI Component Recognition, GUI completion |
| **Papt** | Phone, Tablet | 20,070 | 11,186 | Jan. 2023 | UI Component Recognition, GUI completion, GUI conversion, UI search |

**Broader Applicable Tasks** Table 1 shows a summary of our and other GUI datasets. First, since our data consist of phone-tablet pairs, we must locate the corresponding GUI pages between phones and tablets, resulting in a lesser number of pages than comparable datasets. However, we now have a broader data source (including tablet GUIs), more supported tasks, and newer data. Notably, it is the only available GUI dataset that contains phone-tablet pairwise GUIs, which provides effective data support for the application of deep learning techniques in GUI generation, recommendation, testing, vulnerability detection, and other domains [27, 29, 28, 30, 14].

**Data Accuracy** Specifically, our data eliminates a large number of GUI visual mismatches that are frequent in current datasets like as Rico and Enrico. Due to the limitations of the previous data collection tools, some GUIs have visual mismatches in the metadata and screenshots. Figure 6 shows typical visual mismatch examples between hierarchy metadata and screenshots in current datasets. Based on the bounding box coordinates of Android views provided in the metadata, we depict the location of the views in the metadata as a black dashed line in the screenshot. We mark the obvious visual mismatches with a solid red line box, which do not correspond to any of the views in the rendered screenshot. The metadata provides information on the UI elements behind the current layer, but these elements cannot be interacted with on the current screenshot. Visual mismatches between the UI data in the metadata and the screenshot would result in the UI data in the metadata and the screenshot not corresponding one to the other. Too many mismatch cases would have a negative impact on the efficiency of model generation and search. The selected UI collecting tool, UIautomator2, has optimised the GUI caption technique to avoid metadata and screenshots from containing inconsistent UI information [51]. During manual pair reviews, our volunteers also eliminated GUI pages with mismatched. Compared to other datasets, such as rico, fewer mismatches give us a higher accuracy of our data.

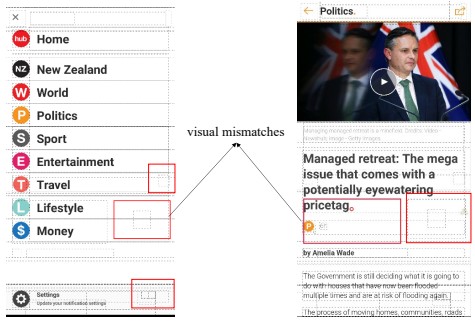

Figure 6: Examples of visual mismatches in current GUI datasets.

# 5 Preliminary Experiments

To demonstrate the usability of our dataset, we perform preliminary experiments on the dataset. Owing to content limitations, we primarily focus on presenting the results derived from the GUI conversion experiment within the main body of the paper. We select current state-of-art approaches for this task and use the automatic metrics for evaluation. We will discuss the qualitative and quantitative performance as well as the limitations of selected approaches on our proposed dataset.

In our supplementary appendix, we have included more experimental results of diverse tasks. Additionally, we provide a more detailed introduction of the employed methodologies, the metrics, and case studies for enhanced comprehension.

## 5.1 Selected Approaches

The GUI conversion task is to automatically convert an existing GUI to a new layout of GUI [8]. For example, given the present phone layout, generate a tablet layout automatically. As far as we know, no researcher has proposed a method for generating a tablet GUI from a phone GUI, so we will use some existing relevant GUI generation methods to apply to this task. Three approaches, which are all widely used in GUI generation and introduced in Section 2, are selected in this task: **LayoutTransformer** [22], **LayoutVAE** [33] and **VTN** [6].

## 5.2 Evaluation Metrics

For GUI conversion task, it's important to evaluate layouts in terms of two perspectives: perceptual quality and diversity. We follow with the similar evaluation protocol in [6, 22, 9] and utilize a set of metrics to evaluate the quality and diversity aspects. Specifically, we use the following metrics:

**Mean Intersection over Union (mIoU) [47]**: also known as the Jaccard index [23], is a method to quantify the percent overlap between the ground-truth and generated output.

**Overlap [54, 22]**: measures the overlap ratio between the ground-truth and our generated output. The overlap metric use the total overlapping area among any two bounding boxes inside the whole page and the average IoU between elements.

**Wasserstein (W) distance [45]**: is the distance of the classes and bounding boxes to the real data. It contains W class and W bbox metrics. Wasserstein distance is to evaluate the diversity between the real and generated data distributions.

**Unique matches [46]**: is the number of unique matchies according to the DocSim [46]. It measures the matching overlap between real sets of layouts and generated layouts. It is designed for diversity evaluation.

**Matched rate**: To more directly show how many UI components in the ground truth's tablet GUI are successfully and automatically converted by the model, we select the metric: the matched rate. Suppose there are a total of $m$ UI components in the ground truth, and there are $n$ components in the generated tablet GUI that match the components in the ground truth, then the matched rate is calculated as $n/m$.

## 5.3 Experimental Setup

Since we are comparing the structure of the rendered GUI as opposed to its pixels, we do not consider the contents of images and texts inside the GUI. Therefore, we convert GUI pages to wireframes by converting each category of UI component to a box of the specified color, which has been widely adopted in recently related works [22, 38, 33]. In our GUI conversion work, unlike typical GUI generation tasks, we request the model to learn to generate a tablet GUI comparable to the input phone GUI, rather than another phone GUI. As input to the training model, we encode all UI components of a phone GUI as a component sequence from top to bottom and left to right. Each UI component in the GUI is encoded as a quaternion $(x, y, w, h)$. Where $x$ and $y$ denote the x and y coordinates of the upper left corner of this UI component, and $w$ and $h$ denote the length and width of the component. Similarly, the tablet GUIs in the pair are encoded into a sequence that serves as the ground truth for training and validation.

Following the setup of experiments in LayoutTransoformer and LayoutVAE, we randomly select 1000 pairs (10%) in the total dataset as the test set and the rest of the data as the training set. All the results of the metrics are calculated on the test set. All of the results are based on 5-fold cross validation on the test set.

## 5.4 Quantitative Evaluation

Table 2: Automatic evaluation results on the test set.

| Model | mIoU ↓ | Overlap ↓ | W class ↓ | W bbox ↓ | # Unique matchces ↑ | Matched rate ↓ |
|---|---|---|---|---|---|---|
| LayoutVAE | 0.10 | 0.23 | 0.29 | 0.012 | 356 | 0.13 |
| LayoutTransformer | 0.12 | 0.32 | 0.31 | 0.024 | 445 | 0.15 |
| VTN | 0.13 | 0.35 | 0.37 | 0.026 | 541 | 0.19 |

We present the quantitative evaluation results in Table 2. As expected, we observe that VTN model achieves the best performances in terms of all metrics. It yields large improvement regarding perceptual quality, such as *W class*, and *Unique matches*. Besides, it obviously outperforms the LayoutVAE model across all metrics and is slightly better than LayoutTransformer model. The reason heavily rely on the mutual enhancement between self-attention and VAE mechanisms. However, according to the final metric *Matched rate*, only less than 20% of the UI components can be matched between the generated tablet GUI layouts and ground truth. Put succinctly, equivalent results can be attained by directly transferring certain phone GUI patterns to the synthesized tablet GUI. The results suggest that further work is needed to design and train a more effective model in the GUI conversion task. We hope that our open-source dataset will enable more researchers to participate in automated UI development and contribute more effective methods.

## 5.5 Qualitative Evaluation

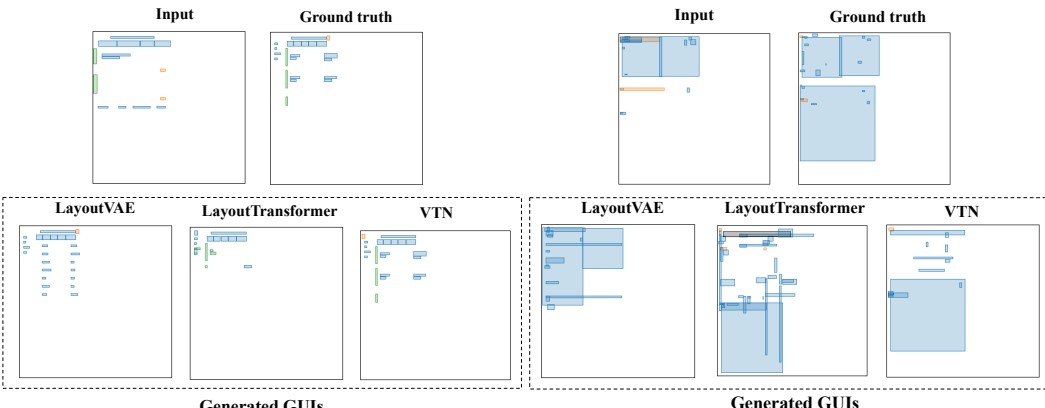

Figure 7: Two examples of generated GUIs by selected approaches

To better understand the performance of different models on our task, we present qualitative results of selected models and their generated outputs in Figure 7. In alignment with the quantitative results in Table 2, we observe that the VTN model outperform other selected models. It demonstrates the efficiency of self-attention mechanism on the layout generation task. However, we can observe that the best selected model VTN still fall short of generating precise margins and positions towards the ground-truth. There is still much room for improvement in learning the relationship between phone and tablet GUIs, and the current results are far from helpful to GUI developers. The GUI conversion in the crossing platform poses huge challenges for the existing state-of-the-art model. Therefore, simply utilizing the previous layout generation model cannot tackle the challenges of GUI conversion in both Android phones and tablets.

## 5.6 Research Questions and Observations from Experiments

In the evaluations delineated in Sections 5.4 and 5.5, it becomes evident that contemporary models fall short in facilitating effective automated GUI development. As illustrated in Table 2, even the

best-performing model, VTN, manages to accurately reproduce a mere 20% of the UI components present in the ground truth of the corresponding table. Notably, the left example in Figure 7 depicts a scenario where our proposed method exhibits a relatively superior GUI generation. However, the instances on the right are notably subpar. We hypothesize that this disparity arises due to the intricate containment and nesting relationships among UI components in the right example, resulting in a more convoluted metadata structure in comparison to the left. Consequently, discerning the relationships between UI components proves challenging for the model.

From our preliminary experiments and insights, we highlight key research queries that currently permeate the domain, aiming to advance the field:

**GUI Encoding and Representation**: Our paper delves into the intricate nature of GUI, a complex data structure encompassing a wide variety of structural information and metadata. The methods of representing and encoding these GUI data are instrumental in facilitating tasks such as data dimensionality reduction and multimodal data fusion, potentially leading to a significant enhancement in the generation effect. This complex encoding presents a ripe area for further exploration and potential innovation.

**Modeling UI relationship**: Furthermore, the logical relationship between UI components, such as those images arranged side by side or images with texts below, provides an exciting avenue for research. Defining and modeling these relationships could pave the way for a new understanding of the interplay between UI components and could have broader applications in automated GUI development.

**Addressing Non-correspondence Contents**: The fragmentation of Android screens poses challenges, particularly when a tablet GUI in a phone-tablet pair houses more content than its phone counterpart. Navigating these content discrepancies presents a valuable research trajectory, with implications for enhancing cross-device GUI adaptability.

## 6  Conclusion

In this paper, we introduce the first pairwise GUI dataset Papt between Android phones and tablets and aims to bridge phone and tablet GUIs. We introduce the characteristics and preliminary experiments in this dataset. We discuss some valuable research problems based on current experimental results and hope our dataset can facilitate the development of deep learning in automated GUI development.

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
