# Papt: A Pairwise GUI Dataset between Android Phones and Tablets

## 1   Dataset Ethics and Responsible Usage

In our commitment to promote responsible research practices, we present a comprehensive set of ethical guidelines for the usage of the Papt dataset. Researchers and developers are urged to be fully aware of these directives and adhere to them in their entirety.

**Intended Use**: The Papt dataset is designed and released strictly for non-profit research purposes. Any commercial use or application is strictly prohibited.

**No Verbatim Publication**: While the dataset provides insights into the GUI layout of various applications, it may contain content from apps that researchers inadvertently downloaded. It is crucial to understand and respect that these contents may be protected under copyright laws. Therefore, users of the Papt dataset must not publish any verbatim content from the applications, such as articles from recognized outlets like BBC news.

**Respect for Copyrights**: The dataset might encompass copyrighted material inherent in some apps' GUI. Users should be aware of potential copyright infringements and must not reproduce, distribute, or create derivative works based on these copyrighted components without necessary permissions.

**Data Responsibility**: While leveraging the dataset for research, users should be cautious not to misuse any part of the data, ensuring that their research does not infringe upon the rights of original content creators or the privacy of users.

**Citing the Dataset**: Any research work that benefits from the Papt dataset should duly acknowledge it. Proper citation ensures that the creators and contributors receive appropriate credit, fostering a community of shared resources and collaborative research.

**Report Violations**: Users are encouraged to report any violations or potential misuse of the Papt dataset. This proactive step will help safeguard the ethical principles that underscore this dataset and the broader research community.

We trust that these guidelines will be meticulously followed, ensuring that the Papt dataset remains a valuable, responsible, and ethical resource for the research community.

## 2   Applications and Impact of the Pairwise GUI Dataset in Research and Industry

A pairwise GUI dataset encompassing both phone and tablet platforms constitutes a pivotal asset in the burgeoning field of automated GUI development. Analyzing this dataset allows for the extraction of underlying patterns, verification of existing methodologies, and the potential training of models to recognize or generate analogous structures. Such an extensive dataset fosters innovation in several crucial domains, including but not limited to cross-platform design, user experience optimization, rigorous testing and validation protocols, adaptive design strategies, and the enhancement of accessi-

Submitted to the 37th Conference on Neural Information Processing Systems (NeurIPS 2023) Track on Datasets and Benchmarks. Do not distribute.

bility features. Beyond merely serving practical developmental needs, this dataset creates a fertile landscape for theoretical exploration, extending the current horizons of modern interface design research.

**Facilitating Cross-Platform Design**: A pairwise GUI dataset that includes representations for both phone and tablet interfaces provides a comprehensive understanding of how GUI elements adapt across different devices. This can enable researchers and developers to create algorithms that facilitate automatic resizing and rearranging of UI components to fit different screens, enhancing cross-platform compatibility.

**Improving User Experience**: By analyzing the relationships between phone and tablet GUI pairs, researchers can identify optimal design patterns that offer a seamless user experience across devices. These insights can guide automated GUI development tools to generate interfaces that maintain consistency and usability, whether viewed on a phone or a tablet.

**Enabling Multi-modal Development**: The rich dataset that includes various components and their metadata in GUI pairs allows for exploration into multi-modal interface development. Researchers can investigate how to intelligently adapt GUI components based on user interaction patterns on different devices. This could lead to automated development processes that personalize the interface according to the specific platform.

**Enhancing Testing and Validation**: With a pairwise dataset, automated testing tools can be developed to compare and contrast how a GUI performs and appears on both phones and tablets. This allows for comprehensive validation processes that ensure consistency and functionality across platforms, thereby reducing development time and improving quality.

**Encouraging Adaptive Design Research**: The differences between phone and tablet GUI pairs highlight the need for adaptive design principles. Researchers can leverage this dataset to experiment with algorithms that automatically adjust GUI elements to different orientations, resolutions, and user preferences, fostering more adaptive and responsive design techniques.

**Facilitating Collaboration**: By establishing a common dataset for phone and tablet GUI pairs, collaboration among researchers is enhanced. It creates a standardized platform for development and evaluation, encouraging innovation in automated design tools, methodologies, and principles.

**Handling Complex Design Challenges**: The detailed information within the dataset, such as UI property and GUI hierarchies, allows for sophisticated modeling of complex design scenarios. This can lead to breakthroughs in handling intricate design challenges like non-correspondence between phone and tablet GUIs, leading to more robust and flexible design solutions.

**Boosting Accessibility Research**: The pairwise dataset may help researchers understand how accessibility features should be adjusted across different devices. Automated tools can then be developed to ensure that interfaces are inclusive and meet accessibility standards, whether accessed via a phone or a tablet.

# 3 Descriptive Statistics of the Papt Dataset

Using the methodologies described in the paper, we have gathered 10,035 valid phone-tablet GUI pairs. To provide a detailed insight into our dataset, we statistically analyze these pairs from two main aspects: the distribution of UI view types discussed in subsection 3.2, and the evaluation of GUI similarities between pairs as outlined in subsection 3.3. Subsection 3.4 details the format of the data in our dataset. Lastly, we discuss the merits of our dataset in comparison to existing GUI datasets in subsection **??**.

## 3.1 Source App Pairs

We first crawl 6,456 tablet apps from Google Play. Then we match their corresponding phone apps by their app names and app developers. Finally, we collect 5,593 valid phone-tablet app pairs from

22 app categories. Table 1 shows the top 15 categories of 5,593 app pairs. Due to the effect of the data's long tail disctribution, we only display the top 15 categories. The column *Category* represents the category of these apps. The column *#Count* and *P(%)* denote the number of apps in this category and their percentage of the overall number of apps, respectively. These 5,593 phone-tablet app pairs are the data source for this dataset. The three most common categories of apps in the data source are: *Entertainment* (8.87%), *Social* (7.04%) and *Communication* (5.83%). As shown in Table 1, the categories of apps in our data source are scattered and balanced. Most of the categories occupy between 4% and 6% of the total dataset. This balanced distribution ensures the dataset's generalizability and diversity.

Table 1: Top 15 categories of source apps.

| Category | #Count | P (%) |
|---|---|---|
| Entertainment | 496 | 8.87 |
| Social | 394 | 7.04 |
| Communication | 326 | 5.83 |
| Lifestyle | 318 | 5.69 |
| Books & Reference | 286 | 5.11 |
| Education | 279 | 4.98 |
| News & Magazines | 271 | 4.85 |
| Shopping | 270 | 4.83 |
| Sports | 267 | 4.78 |
| Music & Audio | 266 | 4.76 |
| Weather | 265 | 4.73 |
| Finance | 262 | 4.68 |
| Bussiness | 261 | 4.67 |
| Travel & Local | 255 | 4.57 |
| Medical | 254 | 4.54 |

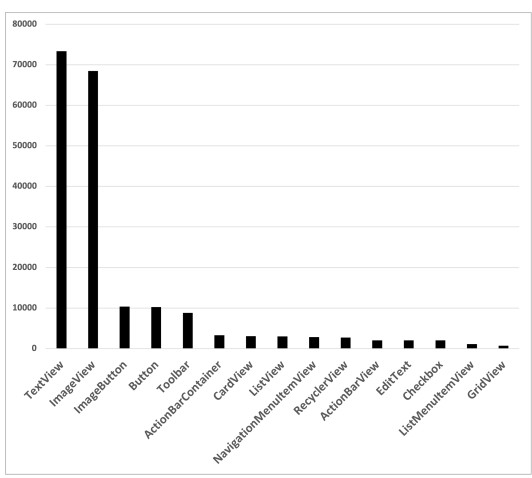

Figure 1: Distribution of top 15 UI view types in the dataset.

## 3.2 Distribution of UI View Types

In Android development, a UI view is a basic building block for creating user interfaces. Views are responsible for drawing and handling user interactions for a portion of the screen [1]. For example, a button, a text , an image, and a list are all a type of view.

Figure 1 illustrates the distribution of UI View types in the dataset. Considering the data's long-tail distribution, we only display the top 15 types. We can see that the *TextView* and *ImageView* types, including all their derived categories such as *AppCompatTextView*, *AppCompatCheckedTextView*, and *AppCompatImageView*, are the most common UI view types in the dataset. Their numbers (73,349 and 68,496) significantly outnumber all other view types. The GUI primarily presents information via text and images, so text and image-related views are the most prevalent in the database. *ImageButton* (10,366) and *Button* (10,235) are the third and fourth most UI views. Users interact with the GUI mainly through clicks and click operations rely heavily on button views, so button-related views are also common in GUI datasets.

## 3.3 Distribution of GUI Pair Similarity

The similarity analysis between phone-tablet GUI pairs is important for downstream tasks. Given a GUI pair, there are a total of $M$ and $N$ GUI views in the GUIs of the phone and tablet, respectively. Suppose there are $L$ the same views in the GUIs of the phone and the tablet. The similarity of their GUIs is calculated as

$$Sim(M, N) = \frac{2 * L}{M + N} \tag{1}$$

Figure 2 shows the frequency histogram of GUI similarities of our phone-tablet GUI pairs in the dataset. The similarity between the GUIs of phones and tablets in most pairs is between 0.5 and 0.7. Considering the difference in screen size between tablets and phones, the current phone GUI page

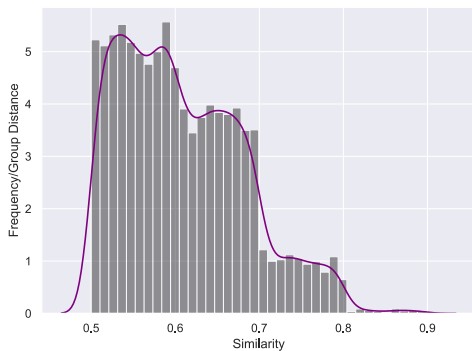

Figure 2: The frequency histogram of GUI similarity of collected pairs

can only contain part of the UI views in the corresponding tablet GUI page, and the current data reminds us that when performing downstream tasks such as GUI layout generation, search, etc., we should consider filling in the contents that are not available in the mobile phone GUI page.

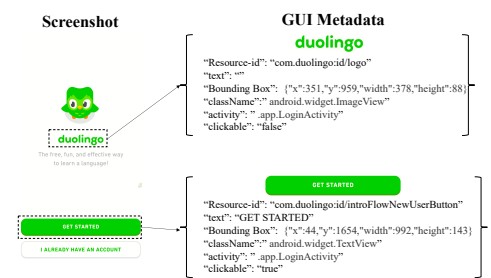

Figure 3: A screenshot of a GUI and its part UI metadata

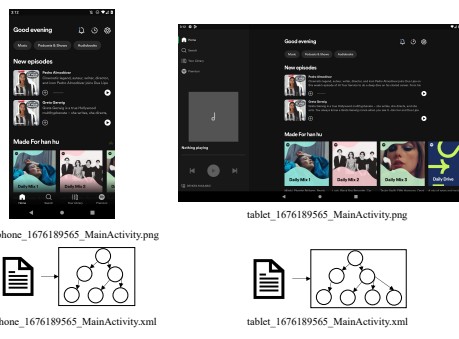

Figure 4: An example GUI pair in Spotify

## 3.4 Data Format

In this section, we first show an example screenshot of a GUI and its corresponding UI metadata. Then, we introduce the format of each GUI pair in the dataset.

**GUI Screenshot and its Metadata** We install and run phone-tablet app pairs in Section **??** on the Pixel6 and Samsung Galaxy tab S8, respectively. We use uiautomator2 [19] to collect screenshots and GUI metadata of the dynamically running apps. Figure 3 shows an example of a collected GUI screenshot and metadata of some UI components inside the GUI. This example is from the app 'Duolingo [9]. The metadata is a documentary object model (DOM) tree of current GUIs, which includes the hierarchy and properties (e.g., class, bounding box, layout) of UI components. We can infer the GUI hierarchy from the DOM tree hierarchy in metadata.

**GUI Pair** Figure 4 shows an example of pairwise GUI pages of the app 'Spotify' in our dataset. All GUI pairs in one phone-tablet app pair are placed in the same directory. Each pair consists of four elements: a screenshot of the GUI running on the phone (*phone_1676189565_MainActivity.png*), the metadata data corresponding to the GUI screenshot on the phone ( *phone_1676189565_MainActivity.xml* ), a screenshot of the GUI running on the tablet (*tablet_1676189565_MainActivity.png* ), and the metadata data corresponding to the GUI screenshot on the tablet (*tablet_1676189565_MainActivity.xml* ). The naming format for all files in the dataset is *Device_Timestamp_Activity Name*. As shown in Figure 4, The filename *tablet_1676189565_-MainActivity.xml* indicates that this file was obtained by the tablet and was collected with the timestamp *1676189565*, this GUI belongs to *MainActivity* and this file is a metadata file in XML format. We use timestamps and activity names to distinguish phone-tablet GUI pairs.

**Pairs in the Dataset** The dataset is made accessible to the public in accordance with the criteria outlined in the attached license agreements[1]. The pairs in the dataset are contained in separate folders according to the app. Most of the app folders are named after the package name of the app's APK, for example, *air.com.myheritage.mobile* , and a few are named after the app's name, for example, *Spotify*. In each app folder, as described in Section 3.4, each pair contains four elements: the phone GUI screenshot, the XML file of the phone GUI metadata, the corresponding tablet GUI screenshot, and the XML file of the tablet GUI metadata. We also shared the script for loading all GUI pairs in the open source repository.

## 3.5 Data Collection Tool

Based on the above-described two collection strategies, we develop two distinct collecting tools: the adjust resolution collector and the similarity matching collector.

The first tool dynamically adjusts the resolution of the current device using ADB instructions. When the running app detects a change in the screen's resolution, it will call the layout file designed for the tablet and change the layout of the current GUI.

The second tool concurrently runs two apps of one app pair on a mobile phone and a tablet. The tool dynamically evaluates the similarity of the GUIs presented on two devices, and automatically collects the matched GUI page pair when the similarity exceeds a predetermined threshold.

These two data collection tools are also included in the repository of the publicly accessible dataset. With the installation instructions provided, more researchers can utilise our tools to collect more customised GUI datasets for future research.

## 3.6 Accessing the Dataset

The dataset is made accessible to the public in accordance with the criteria outlined in the attached license agreements[2]. The pairs in the dataset are contained in separate folders according to the app. Most of the app folders are named after the package name of the app's APK, for example, *air.com.myheritage.mobile* , and a few are named after the app's name, for example, *Spotify*. In each app folder, each pair contains four elements: the phone GUI screenshot, the XML file of the phone GUI metadata, the corresponding tablet GUI screenshot, and the XML file of the tablet GUI metadata. We also shared the script for loading all GUI pairs in the open source repository.

# 4 Preliminary Experiments

To demonstrate the usability of our dataset, we perform preliminary experiments on the dataset. These experiments contain two types of tasks: GUI conversion and GUI retrieval. We select current state-of-art approaches for these two tasks and use the automatic metrics for evaluation. We will discuss the qualitative and quantitative performance as well as the limitations of selected approaches on our proposed dataset.

## 4.1 Tasks

**GUI Conversion** This task is to automatically convert an existing GUI to a new layout of GUI [3]. For example, given the present phone layout, generate a tablet layout automatically. It is also a GUI generation task. The goal of GUI conversion task is to generate a flexible interface that is applicable to different platform. By generating GUI layouts that fit the needs of developers, we can make developer more productive and decrease engineering effort.

---

[1]`https://github.com/huhanGitHub/papt`
[2]`https://github.com/huhanGitHub/papt`

**GUI Retrieval** This task is to retrieval the most relevant GUI from the database and recommend it to the most appropriate users [21]. GUI template search and recommendation is an essential direction for current automated GUI development to accelerate the development process.

## 4.2 Selected Approaches

### 4.2.1 GUI Conversion

As far as we know, no researcher has proposed a method for generating a tablet GUI from a phone GUI, so we will use some existing relevant GUI generation methods to apply to this task. Three approaches, which are all widely used in GUI generation, are selected in this task:

**LayoutTransformer** [10]: a simple yet powerful auto-regressive model based on Transformer framework that leverages self-attention to generate layouts by learning contextual relationships between different layout elements. It is able to generate a brand new layout either from an empty set or from an initial seed set of primitives, and can easily scale to support an arbitrary of primitives per layout.

**LayoutVAE** [13]: a stochastic model based on variational autoencoder architecture. It is composed of two modules: CountVAE which predicts the number of objects and BBoxVAE which predicts the bounding box of each object. It is capable of generating full image layouts given a label set, or per label layouts for an existing image given a new label. Besides, it is also capable of detecting unusual layouts, potentially providing a way to evaluate layout generation problem.

**VTN** [2]: a VAE-based framework advanced by Transformer model, which is able to learn margins, alignments and other elements without explicit supervision. Specifically, the encoder and decoder are parameterized by attention-based neural network. During the variational process, VTN sample latent representations from the prior distributions and transform those into layouts using self-attention based decoder.

### 4.2.2 GUI Retrieval

Researchers have not yet attempted to get the comparable tablet GUI design from a phone GUI design, but numerous methods for locating relevant GUI designs have been proposed. These comparable GUI design retrieval approaches will be selected and adapted to fit this task. For GUI retrieval task, we employ three semantic matching and learning-based models:

**Rico** [8]: a neural-based training framework that utilizing content-agnositc similarity heuristic method for UI comparing and matching. To facilitate query-by-example search, Rico exposes a vector representation for each UI that encodes layout. Rico provides search engines with several visual representations that can be served up as results: UI screenshots, flows, and animations.

**GUIFetch** [4]: a method that takes an input the sketch for an app and leverages the growing number of open source apps in public repositories and then devise a component-matching model to rank the identified apps using a combination of static and dynamic analysie and computes a similarity metric between the models and the provided sketch.

**WAE** [7]: a wireframe-based UI searching model using image autoencoder architecture. Specifically, it is a neural based approach using convolutional neural network (CNN) in an unsupervised manner for building a UI design search engine that is flexible and robust in face of the great variations in UI designs. The enhancement of wireframe will facilitate the layout generation process.

Table 2: Automatic evaluation results on the test set.

| Model | mIoU ↓ | Overlap ↓ | W class ↓ | W bbox ↓ | # Unique matchces ↑ | Matched rate ↓ |
|---|---|---|---|---|---|---|
| LayoutVAE | 0.10 | 0.23 | 0.29 | 0.012 | 356 | 0.13 |
| LayoutTransformer | 0.12 | 0.32 | 0.31 | 0.024 | 445 | 0.15 |
| VTN | 0.13 | 0.35 | 0.37 | 0.026 | 541 | 0.19 |

### 4.3 Evaluation Metrics

We consider automatic evaluation on both these two tasks.

#### 4.3.1 GUI Conversion

For GUI conversion task, it's important to evaluate layouts in terms of two perspectives: perceptual quality and diversity. We follow with the similar evaluation protocol in [2, 10, 5] and utilize a set of metrics to evaluate the quality and diversity aspects. Specifically, we use the following metrics:

**Mean Intersection over Union (mIoU) [18]**: also known as the Jaccard index [11], is a method to quantify the percent overlap between the ground-truth and generated output. The IoU is calculated by dividing the overlap area between predicted class positions and ground truth by the area of union between predicted position and ground truth. So, it is computed by

$$mIoU = \frac{1}{k}\sum_{i=0}^{k}\frac{TP(i)}{TP(i) + FP(i) + FN(i)} \tag{2}$$

where $k$ means $k$ classes in both images, $TP(i)$, $FP(i)$ and $FN(i)$ represent the distribution of true positive, false positive and false negative of $i_{th}$ class between two compared images.

**Overlap [20, 10]**: measures the overlap ratio between the ground-truth and our generated output. The overlap metric use the total overlapping area among any two bounding boxes inside the whole page and the average IoU between elements.

**Wasserstein (W) distance [15]**: is the distance of the classes and bounding boxes to the real data. It contains W class and W bbox metrics. Wasserstein distance is to evaluate the diversity between the real and generated data distributions.

**Unique matches [16]**: is the number of unique matchies according to the DocSim [16]. It measures the matching overlap between real sets of layouts and generated layouts. It is designed for diversity evaluation.

**Matched rate**: To more directly show how many UI components in the ground truth's tablet GUI are successfully and automatically converted by the model, we select the metric: the matched rate. Suppose there are a total of $m$ UI components in the ground truth, and there are $n$ components in the generated tablet GUI that match the components in the ground truth, then the matched rate is calculated as $n/m$.

#### 4.3.2 GUI Search

For GUI retrieval task, we evaluate the performance of a UI-design search method by two metrics: Precision and Mean Reciprocal Rank (MRR), which have been widely-used in GUI search [7, 8, 21, 6, 12].

**Precision@k (Pre@k)**: Precision@k is the proportion of the top-k results for a query UI that are relevant UI designs. Specifically, the calculation of ranking Precision@k is defined as follows:

$$\text{Precision@k} = \frac{\#relevantUIdesign}{k}, \tag{3}$$

As we consider the original UI as the only relevant UI for a treated UI in this study, we use the strictest metric Pre@1: Pre@1=1 if the first returned UI is the original UI, otherwise Pre@1=0.

**Mean Reciprocal Rank (MRR)**: MRR is another method to evaluate systems that return a ranked list. It computes the mean of the reciprocal rank (i.e., 1/rank) of the first relevant UI design in the search results over all query UIs. Specifically, the calculation of MRR is defined as follows:

$$MRR = \frac{1}{Q}\sum_{i=1}^{Q}\frac{1}{rank_i}, \tag{4}$$

where $Q$ refers to the number of all the query UIs. The higher value a metric is, the better a search method performs.

### 4.4 GUI Conversion Task

#### 4.4.1 Experimental Setup

Since we are comparing the structure of the rendered GUI as opposed to its pixels, we do not consider the contents of images and texts inside the GUI. Therefore, we convert GUI pages to wireframes by converting each category of UI component to a box of the specified color, which has been widely adopted in recently related works [10, 14, 13]. In our GUI conversion work, unlike typical GUI generation tasks, we ask the model to learn to generate a tablet GUI comparable to the input phone GUI, rather than another phone GUI. As input to the training model, we encode all UI components of a phone GUI as a component sequence from top to bottom and left to right. Each UI component in the GUI is encoded as a quaternion $(x, y, w, h)$. Where $x$ and $y$ denote the x and y coordinates of the upper left corner of this UI component, and $w$ and $h$ denote the length and width of the component. Similarly, the tablet GUIs in the pair are encoded into a sequence that serves as the ground truth for training and validation.

Following the setup of experiments in LayoutTransoformer and LayoutVAE, we randomly select 1000 pairs ( 10%) in the total dataset as the test set and the rest of the data as the training set. All the results of the metrics are calculated on the test set. All of the results are based on 5-fold cross validation on the test set.

#### 4.4.2 Quantitative Evaluation

We present the quantitative evaluation results on the Table 2. As expected, we observe that VTN model achieves the best performances in terms of all metrics. It yields large improvement regarding perceptual quality, such as *W class*, and *Unique matches*. Besides, it obviously outperforms the LayoutVAE model across all metrics and is slightly better than LayoutTransformer model. The reason heavily relies on the mutual enhancement between self-attention and VAE mechanisms. However, according to the final metric *Matched rate*, only less than 20% of the UI components can be matched between the generated tablet GUI layouts and ground truth. We demonstrate that most phone-tablet GUI pairs in the dataset have a similarity between 50% and 70%, so the current model's capacity to learn the relationship between phone-tablet GUI pairs still requires improvement. The results suggest that further work is needed to design and train a more effective model in the GUI conversion task. We hope that our open source dataset will enable more researchers to participate in automated UI development and contribute more effective methods.

#### 4.4.3 Qualitative Evaluation

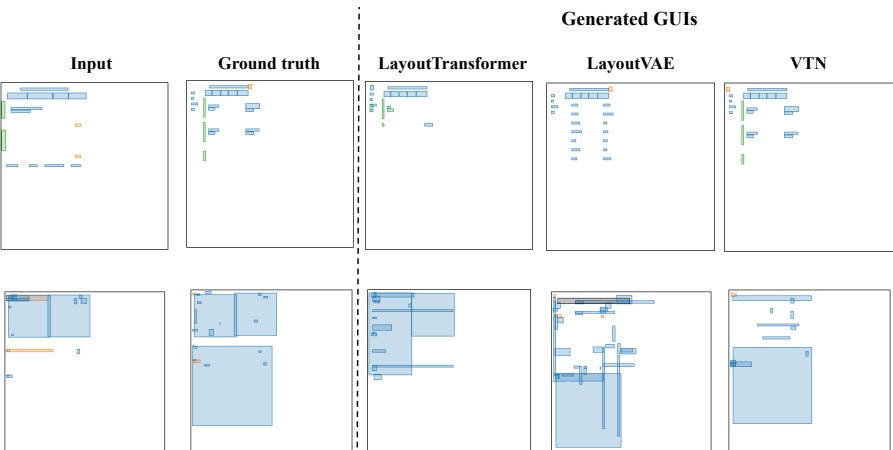

Figure 5: Examples of generated GUIs by selected three approaches

To better understand the performance of different models on our task, we present qualitative results of selected models and their generated outputs in Figure 5. In alignment with the quantitative results

in Table 2, we observe that the VTN model outperform other selected models. It demonstrates the efficiency of self-attention mechanism on the layout generation task. However, we can observe that the best selected model VTN still fall short of generating precise margins and positions towards the ground-truth. There is still much room for improvement in learning the relationship between phone and tablet GUIs, and the current results are far from helpful to GUI developers. The GUI conversion in the crossing platform poses huge challenges for the existing state-of-the-art model. Therefore, simply utilizing previous layout generation model cannot tackle the challenges of GUI conversion in both Android phones and tablets.

### 4.5 GUI Retrieval

### 4.5.1 Experimental Setup

Following related GUI search works [7, 17], we convert the GUI pages into wireframes in the same way as the GUI conversion task in Section 4.4.1. The input in this experiment is the phone GUI wireframe and the ground truth is the corresponding tablet GUI wireframe. We train a GUI encoder to encode GUI into numerous vectors and compare the similarity of these numerous vectors. We randomly select 1000 pairs ( 10%) in the total dataset as the test set and the rest of the data as the training set. The results of this test set is automatically checked.

Due to the existence of comparable GUI designs on the tablet, sometimes the search result is not the ground truth, but the GUI design is reasonable and very close to the ground truth. We would also consider this result to be an appropriate design. Therefore, we randomly selected 100 cases from the test set and manually verify the results. We invited three industry developers with at least one year of experience in Android development to manual evaluate search results. Each participant evaluate the top 10 search results to determine if they are a reasonable tablet design. After the initial evaluating, three volunteers have a discussion and merge conflicts. They clarify their findings, scope boundaries among categories and misunderstanding in this step. Finally, they iterate to revise evaluation results and discuss with each other until consensus is reached. To differentiate, we call these results manual checked results.

Table 3: Qualtitative results for GUI retrieval and recommendation task.

| Model | Pre@1 | Pre@5 | Pre@10 | MRR |
|---|---|---|---|---|
| Rico [8] | 0.73 | 0.65 | 0.58 | 0.69 |
| GUIFetch [4] | 0.63 | 0.54 | 0.52 | 0.62 |
| WAE [7] | 0.80 | 0.77 | 0.75 | 0.83 |

### 4.5.2 Results

Table 3 shows the performance metrics of the three selected methods in our dataset. We report four metrics in terms of precision following with previous UI search work: *Pre1*, *Pre5*, *Pre10* and *MRR*. The results in column *Pre@1* and *MRR* are automatic checked results. The results in column *Pre@5* and *Pre@10* are manual checked results. We can observe that the *WAE* model outperforms other two approaches in terms of automated and manual results. Compared with GUI conversion task, GUI search task is more developed and applicable. Learning-based approaches *Rico* and *WAE* both achieved high search accuracy. It demonstrates the advantages of neural network models with huge training parameters in extracting features and patterns of GUI. We hope that more researchers in the future will design more search-related tasks based on our dataset, such as semantic-based search, GUI template recommendation, etc.