# OpenReview forum: "Pairwise GUI Dataset Construction Between Android Phones and Tablets"
_NeurIPS.cc/2023/Track/Datasets_and_Benchmarks — NeurIPS 2023 Datasets and Benchmarks Poster_

### Official Review · Reviewer_ZrJu · 2023-07-21
**Papt: A Pairwise GUI Dataset between Android Phones and Tablets**

**Rating:** 4
**Confidence:** 4
**Correctness:** The dataset is too small, and only co…
**Clarity:** Yes, the paper is well-written and ea…

**Strengths:**

1. The pairwise GUI dataset between phones and tablets is interesting and critical.
2. The authors also provide the way to collect more data.

**Additional Feedback:**

Please see the weakness and limitations above.

**Documentation:**

The documentation is not provided, only the UIAutomator2 Github Repo are provided.
Also, the dataset in the Github only contains a few samples.

**Ethics:**

No.

**Limitations:**

1. The proposed dataset can somehow address the limitations in the GUI related problems, but it only contains a small number of samples under limited scope and system.
2. The experimental results are expected to provide more insights, for example, what is the major research question in this research field?

**Opportunities For Improvement:**

1. The size of dataset is quite tiny, which lacks broader impact.
2. The dataset only contains Android applications, which may not generalize to the iOS applications.
3. The experimental findings are not sufficient for general research problem definition.

**Relation To Prior Work:**

The dataset is actually new. However, the collection process is somehow easy to realize, and many existing automated testing tools can also provide the pairwise dataset.

**Summary And Contributions:**

This paper propose a pairwise dataset for phone-tablet GUI conversion and retrival. The authors describe the dataset collection, statistics and conduct empirical experiments on the provided datasets.

---

> ### Author Response · Authors · 2023-08-13
> **Clarification and discussion of dataset size, dataset accessibility and related tools**
>
> Dear reviewer,
>
> We would like to extend our sincere gratitude for the time and effort you have put into reviewing our manuscript. Your insightful comments are valuable to us, and we appreciate the opportunity to address the concerns you have raised.
> Upon reading your review, we realized that our expression in the paper might have led to some misunderstandings. These factual errors may stem from our poor explanation, and we appreciate your bringing them to our attention. Please allow us to clarify the following points:
>
> ### (1) Dataset Size
> The dataset comprising 10,035 GUI pairs is, in fact, not tiny, and we appreciate your attention to this critical aspect of our research. We'd like to delve further into why this dataset is not tiny and so essential:
>
> **Complexity and Diversity**:
> Unlike text and image data, each phone GUI contains more than 10 UI components and its corresponding metadata in a GUI pair. As shown in Figure 4 of the paper, each pair provides a rich and diverse resource. This includes the GUI layout, UI property, and GUI hierarchies, making it an intricate and multifaceted collection. Other reviewers (abm3 and BuL6) have acknowledged the size of the dataset as being quite large, and we regret if our description did not adequately convey this. We've revised the introduction part to emphasize the dataset size more explicitly.
>
> **Challenges in Data Collection**:
> The number of tablet-specific apps on Google Play is limited compared to phone apps, making data collection challenging. Curating such a dataset is labor-intensive, necessitating careful planning, execution, and validation to ensure its representativeness and utility.
>
> **Novelty and Contribution**:
> Our provision of a pairwise dataset is both novel and a significant contribution to the research community. We are the first to address the distinct GUI design challenges for phones and tablets in this domain. This pioneering effort has required extensive research and innovation, involving interdisciplinary knowledge and practical techniques.
>
> We understand that the magnitude of our dataset might be seen differently based on various perspectives. Still, we sincerely believe that these clarifications provide a more comprehensive view of its importance and novelty. We are committed to continuous improvement and welcome further dialogue on this matter.
>
> ### (2) Data Collection Tools:
> Our innovative approach in pairwise GUI data collection distinguishes our work from traditional methodologies. Unlike prevalent testing tools focused on single GUIs, our unique capability is in gathering pairwise GUIs—a previously unaddressed gap in the field. As highlighted in our introduction and related work sections, we haven't identified mainstream tools that offer this feature, emphasizing our contribution's novelty. If we've inadvertently overlooked any and possible, we'd appreciate any insights or tool recommendations from the reviewer to further enrich our study.
>
> ### (3) Platform Specificity:
> While it is true that our dataset in the current paper focuses predominantly on Android apps, the novelty and contribution of our work also contain pairwise data collection method. This method, though Android-centric, is adaptable for iOS apps. Given similar interactions and screen sizes between iOS and Android, many layout designs are analogous, suggesting that insights from our Android dataset can be transferable to the iOS ecosystem.
>
> We acknowledge the absence of iOS GUI data as a potential direction for further exploration. However, we respectfully posit that this limitation does not undermine the core value and innovations presented in the current paper. Instead, it provides an avenue for future endeavors, and should not be perceived as a critical shortcoming of our current efforts.
>
>
> ### (4) Current Major Research Questions:
> Thank you for emphasizing the need for deeper analysis of prevailing research questions in our domain. While space constraints initially limited our exploration, we've now expanded on this topic in section 5.6 of our paper, discussing GUI Encoding, Modeling UI relationships, and Non-correspondence Content challenges. We trust that these additions better align our work with overarching field trends and hope they provide a more holistic perspective for readers.
>
> ### (5) Dataset Accessibility:
> We have supplied a link to the entire dataset and included documentation in the GitHub readme, as highlighted by Reviewers abm3, s1Us, and BuL6. The dataset's omission from GitHub is a result of file size limitations on the platform, rather than an oversight. In response to your feedback, we have updated our GitHub readme to enhance clarity on this matter.
>
> In conclusion, we are deeply committed to addressing your concerns and revised our manuscript to provide clearer explanations where needed. We understand the importance of clear communication and hope that these clarifications dispel any misconceptions.

---

### Official Review · Reviewer_BuL6 · 2023-07-21
**Papt: A Pairwise GUI Dataset between Android Phones and Tablets**

**Rating:** 6
**Confidence:** 3
**Correctness:** Yes, the dataset has been collected i…
**Clarity:** The paper is well written

**Strengths:**

1. The dataset is large
2. The authors have manually verified this dataset by a set of application developers
3. They discuss why prior dataset might have erroneous data

**Additional Feedback:**

The authors should clearly tell how such a paired dataset would be helpful for future research

**Documentation:**

Yes

**Ethics:**

Yes

**Limitations:**

1. The previous dataset was for only one platform and the authors collected it for two platforms. Apart from that they got it validated via a set of application developers. Though they don't report how many samples were discarded after validation

**Opportunities For Improvement:**

1. Apart from the dataset collection, there are no significant contributions of the paper. The paper does not clearly mention where and how this dataset would be helpful.
2. The data collection process is described in the supplementary, however, I feel it should have been included in the main paper. The only contribution of this paper is data collection for GUI, hence it should better clearly tell how such a collection was done and what are the challenges for such data collection.
3. The paper follows a decompilation procedure for getting access to the layout file. However, many Android developers create dynamic UI components i.e., through Java/Kotlin source code such as creating a button in the source java/kotlin code not specifying it through the layout XML file. Can this data collection method capture such dynamic content?

**Relation To Prior Work:**

Yes

**Summary And Contributions:**

The paper provides a dataset of GUI pairs of Tablet and mobile phones.  They show that how such a dataset is created and how it can used to auto generate GUI for another platform

---

> ### Author Response · Authors · 2023-08-13
> **Response to Reviewer BuL6**
>
> Dear Reviewer,
>
> We sincerely thank you for your insightful review of our manuscript. Your feedback has been invaluable in identifying strengths and areas for enhancement. We appreciate your recognition of our work and have addressed the concerns raised.
>
> ### (1) where and how this dataset would be helpful for future research.
> Thank you for bringing attention to the perceived lack of elaboration on how our dataset contributes to the field of GUI automated development. We acknowledge this oversight and appreciate your valuable insights.
> In the introduction of our paper, we indeed highlighted the role of GUI automated development and identified the scarcity of a pairwise GUI dataset as a research bottleneck. This was our primary focus, and we aimed to demonstrate the novelty and necessity of our dataset in this area.
>
> However, we recognize that our initial submission may not have adequately communicated the broader applications and significance of our work. As you've rightly pointed out, we need to articulate more clearly where and how this dataset would be helpful to both researchers and developers.
>
> To address your concerns, we have made the following revisions:
>
> **Clarification in Introduction**:
> We've added more details in the last paragraph of the introduction to elaborate on the helpfulness of our dataset in the field of automated GUI development. Specifically, we now emphasize how the pairwise GUI dataset between phone and tablet can lead to innovations in cross-platform design, user experience enhancement, testing and validation, adaptive design, and accessibility. This additional explanation is intended to offer readers a more comprehensive understanding of our dataset's relevance and potential impact.
>
> **Detailed Explanation in Supplementary File**:
> In response to your suggestion, we have added a new section ‘Applications and Impact of the Pairwise GUI Dataset in Research and Industry’ in the supplementary file that provides an in-depth explanation of how our dataset will support and enrich existing research. This section is designed to offer readers who seek more detailed insights a thorough examination of the dataset's applications.
>
>
> We sincerely hope that these adjustments align with your expectations and elevate the overall quality of our paper.
> We remain open to further communication and clarification with you if you have any additional questions or concerns.
>
>
> ### (2) Data collection
> Thank you for your insightful observation regarding the placement of the data collection process within our paper. We recognize the importance of detailing the collection methodology within the main text, especially given that the data collection for GUI represents the central contribution of our work.
>
> Initially, we placed the data collection section in the supplementary materials due to constraints related to the article's length and a desire to maintain focus on the broader themes. However, upon reflection and considering your valuable feedback, we appreciate that a clear and comprehensive understanding of the data collection process, including its challenges, is integral to the overall contribution of the paper.
>
> In response, we have undertaken a thorough restructuring of the paper, optimizing the layout to incorporate the data collection process back into the main text. This reorganization ensures that the data collection methodology, its intricacies, challenges, and the rationale behind our approach, are readily accessible to readers within the core narrative of the paper.
>
> ### (3) Can this data collection method capture such dynamic content?
>
> Thank you for your perceptive observation on the aspect of UI similarity-based GUI pairing. Indeed, as delineated in the 'UI Similarity-Based GUI Pairing' section of our data collection methodology, we addressed the challenge of handling apps that do not pre-set XML layout files for different screens. This includes scenarios such as developing mirror apps or dynamically adapting various screens through Java/Kotlin coding.
>
> Your insight has helped us identify a potential ambiguity in our description that may lead to misunderstandings. We have taken your suggestion to heart, and in the revised manuscript, we have explicitly clarified this aspect in the data collection section. We now clearly articulate that apps lacking platform-specific XML definitions, encompassing those that employ mirror apps or utilize Java/Kotlin code to dynamically adapt to disparate screens, are handled through our innovative UI similarity matching method.
>
> ### (4) how many samples were discarded after manual validation
> Thanks for your suggestion, we reorganize the data and find that we remove 2,296 pairs of logical inconsistency and obvious visual mismatches between screenshots and their metadata. We revise our paper accordingly.
>
> Once again, we're genuinely thankful for your keen observations and hope our revisions align with your suggestions.

---

> > ### Comment · Reviewer_BuL6 · 2023-08-29
> > **Utilizing the dataset for future research**
> >
> > Thanks for your clarification and putting the methodology in the main paper.
> > Though the methodology has been added, 3.1, 3.2, and Figure 2, they are not highlighted in blue. They are new content.
> > Thanks for discussing how such a dataset can be used by future research (by updating the intro in the main paper and supplementary). However, none of the future work discussed has any references, hence it is not clear whether no such work has happened utilizing such a dataset or just missing citations. The only citation provided in intro is [41] regarding future work, which is 2015 archive paper on generative models. It would have been more useful if the authors directly discussed by putting references on how such a pairwise dataset can be utilized.

---

> > > ### Author Response · Authors · 2023-08-30
> > > **respnose to 'Utilizing the dataset for future research'**
> > >
> > > Thank you for your feedback and suggestions. We apologize for not highlighting the new content in blue. We understand that this may have caused inconvenience during your review, and we thank you for bringing this to our attention.
> > > We would like to clarify that the revised contents, including sections 3.1, 3.2, and Figure 2, are included in the supplementary material of the last paper version. Therefore, they were not marked as blue in the main paper. We acknowledge that this lack of clarity may have caused confusion during your review, and we apologize for any inconvenience caused.
> > >
> > > Regarding the discussion on future work, we understand your concern about the lack of references. In the next version of the paper, we will ensure that references are provided to support the discussion on how such a pairwise dataset can be utilized. We value your continued support for our paper and will make the necessary revisions accordingly.
> > > However, due to the limited time available during the discussion period, it is possible that we may not be able to update the manuscript by the rebuttal deadline. Rest assured, we will carefully consider your comments and incorporate them into the paper during the camera-ready phase.
> > >
> > > Thank you once again for your valuable feedback.

---

### Official Review · Reviewer_s1Us · 2023-07-22
**A Potentially Useful Dataset for GUI Layout Generation**

**Rating:** 8
**Confidence:** 4

**Strengths:**

1. This is the first GUI dataset that pairs the UI between the phone and tablet app.

2. The dataset is carefully cleansed to avoid errors.

**Additional Feedback:**

Minor nitpicks:

- Formatting of Footnote 2
- Inconsistency in "UI search" vs "GUI search" in Table 2.


**Clarity:**

The writing is generally straight forward and clear, modulo a few points mentioned in the other parts of the review.

**Correctness:**

The dataset collection procedure seems to be sound and careful, with human in the loop for verification.

**Documentation:**

The dataset is clearly documented in the paper, supplementary PDF, and the GitHub page.

**Ethics:**

Issues about ethics are not discussed in the paper.

The dataset contains screenshots from apps and their content (e.g., BBC).  Copyright issues and adherence to the terms and conditions of the apps should be discussed.

**Limitations:**

There is no limitation section.

The authors are encouraged to consider my comments in the "Opportunities for Improvement".

**Opportunities For Improvement:**

1. Given that pairing between UI elements in the phone and tablet app is the novelty in this dataset, the process of pairing is important and should be elaborated in the main paper, rather than in the supplementary materials.  To make space for this, I suggest the details about distribution of categories, GUI similarities, can be relegated to the supplementary materials.  Including the highlights of the statistics is enough.

2. One part that is unclear to me is what happen when the table apps contain features not available in the phone app.  Is this common in the dataset?  How many pages in the tablet dataset is unmatched in the phont dataset?

3. Another concern is the quality of the apps.  If researchers are going to use this dataset to train and evaluate the quality of generated GUI, it is important that the GUI in this dataset is of good quality (e.g., follows good UI/UX guideline).  The Google Play store is a jungle and it would be useful if only well-rated apps are included.

4. The notion of "same view" is unclear to me.  Sorry if I missed the explanation but if this is not explained, it needs to be.

**Relation To Prior Work:**

The difference between this dataset and prior work is clearly explained.  Table 2 is nice.

p/s: Table 2 should be bigger.  The column on #Paired GUIs is not informative.

**Summary And Contributions:**

This data includes slightly over 10K GUI page pairs, crawled from 5.6K Android apps on phone and tablet.

---

> ### Author Response · Authors · 2023-08-13
> **Response to Reviewer s1Us**
>
> Dear Reviewer,
>
> We would like to express our sincere gratitude for your insightful review of our manuscript. Your feedback has provided valuable guidance, highlighting both the strengths and areas for improvement. We are pleased to see your recognition of the merits of our work, and we take the opportunity to address the concerns you have raised.
> ### (1) Pairing Approach:
> We greatly appreciate your recommendation to emphasize the GUI pairing process in the main paper, given its novelty. We concur that the comprehensive delineation of the data collection process is critical to the paper's foundation.
>
> In our initial draft, the placement of this section in the supplementary materials was driven by paper length constraints and thematic flow considerations. However, echoing your feedback, we've reintegrated this pivotal section into the main text. We've endeavored to strike a balance by incorporating the essentials of the pairing methodology while relegating more granular details like GUI similarities and distribution categories to supplementary materials, as you wisely suggested. We trust that this adjustment augments the manuscript's coherence and contextualizes the significance of our data collection method more prominently.
>
> ### (2) Tablet-Phone Page Discrepancies:
>
> Your query regarding unmatched pages between tablet and phone apps is apt. Given the variable screen real estate, it's commonplace for tablet apps to harbor additional features, especially in content-rich pages like app home screens.
>
> Our methodology tackles this challenge twofold: Firstly, through an automated similarity assessment between GUI pairs, with pairs scoring below a 0.5 similarity threshold being bypassed. Secondly, during our meticulous manual verification, we screen for both visual mismatches and logical coherence between paired GUIs. Where a tablet GUI presents a surfeit of non-matching content, it's deemed non-coherent and discarded. Through this rigorous process, 2,296 out of 12,331 pairs are eliminated due to logical inconsistency and obvious visual mismatches. Although the exact percentage that was discarded due to low logical consistency did not have enough time to be recalculated, our initial assessment suggests it may be in the range of 10-20%.
>
> ### (3) App quality concerns
> Thank you for your concern, this is a very good question. We fully understand your concerns regarding the qualitative aspects of the apps in our dataset. When collecting data, in order to allow us to have a comprehensive observation of the current GUI pairs, we did not intentionally distinguish GUI pairs from the app’s ratings. However, our data collection approach, which targeted popular apps on the Google Play store, indirectly filtered many lesser-quality apps. Nevertheless, given the dataset's organization by app name or package name, we're poised to further refine our dataset, if needed, by correlating with app ratings. On this way, you may alleviate this concern.
>
> ### (4) Clarifying the "Same View" Concept
>
> Our apologies for any oversight in explaining the "same view" notion clearly, we explained in the data collection section how we defined two ui components as the same views, but placed in the supplement. According to your suggestion, we added the definition of same views in section 3.3 of the revised manuscript.
>
> To elaborate:
>
> Herein, UI components are primarily distinguished based on their typologies and inherent properties.
> Components from both mobile and tablet platforms, exhibiting analogous types and attributes, are subsequently deemed as the same views.
>
>
> Moreover, in response to your feedback, we've made table optimizations by removing the '#Paired GUIs' column in table 2 to provide a more concise overview.
>
> Once again, we're genuinely thankful for your keen observations and hope our revisions align with your suggestions. We're committed to further refining our work based on any additional feedback you might have.

---

> > ### Comment · Reviewer_s1Us · 2023-08-29
> > **Acknowledgement**
> >
> > Thank you for your response.  I am happy with the proposed revision to strengthen the paper and my ratings remain unchanged.

---

> > > ### Author Response · Authors · 2023-08-30
> > > **response to 'Acknowledgement'**
> > >
> > > Thank you for your response. We are delighted to hear that you are satisfied with the proposed revision and that your ratings remain unchanged. We are glad that our revision effectively addressed your previous questions and concerns.
> > >
> > > We would like to express our sincere gratitude for your valuable comments and support for our work. Your feedback has been instrumental in improving the quality of our paper, and we appreciate your time and effort in reviewing it.
> > >
> > > Thank you once again for your help.

---

### Official Review · Reviewer_abm3 · 2023-07-24
**Interesting and *potentially* useful dataset for phone/table UI conversion**

**Rating:** 6
**Confidence:** 3
**Correctness:** The paper appears correct.

**Strengths:**

The paper has a clear value proposition: a dataset to train models that automate generation of a table UI from phone UI, and vice versa. This is a practical and valuable problem.

The dataset construction is clever and reasonably rigorous, including manual review. The dataset is sensibly structured, large, and diverse (across types of app).

The paper is well structure: problem, related work, author's dataset, analysis of dataset, training of models on dataset, analysis of results on models. It flows and reads well.



**Additional Feedback:**

Interesting idea, well executed -- would like to see a bit more analysis of model efficacy.

**Clarity:**

The paper is clearly structured and written. It reads well, with some minor typos.

**Documentation:**

The author's Github repo has easy to follow directions and complete collection of useful code.

**Ethics:**

The paper does scrape data from existing apps. This is for research purposes, but is a tricky ethical subject and some treatment of the topic might be warranted in the paper.

**Limitations:**

The authors are direct about the their trained models being insufficient to be practical.

**Opportunities For Improvement:**

The paper is strongest through construction of the dataset, but weakens substantially in describing experiments with using the dataset. By certain metrics, simply copying the UI from one device to another would substantially outperform the model (see similarity in construction vs matched rate in experimentation). This level of performance seems almost cursory, though the task is admittedly complex. However, value in ML is much more grounded in the data than in the model, so this is far from a fatal flaw.

Some specific points:
-- I'd like more detail on the manual review. What did it entail?
-- Figure 7 is important but hard to understand. It's be clearer to separate the phone->table and tablet->phone tasks into two subfigures IMO.
-- I'd really like more analysis of relatively good/bad outcomes for the model -- including maybe a hacky visual mapping from one UI to another (automatic copy and paste with scaling) -- to make it easier to get an intuitive grasp of how close this technique is to useable.





**Relation To Prior Work:**

The authors clearly describe relation to prior work. In particular "Table 2: Comparison between our dataset and other GUI datasets" is something I'd like to see all dataset authors include a variant of.

**Summary And Contributions:**

The paper describes a dataset that of [ phone, tablet ] UI pairs with corresponding UI elements. The dataset is scraped from existing applications with real UI pairs, automatically annotated, and manually verified. The authors attempt to train several basic models on the dataset to convert from one device UI to the other, but their attempts are minimally successful and show need for further work.

---

> ### Author Response · Authors · 2023-08-13
> **Response to Reviewer abm3**
>
> Dear Reviewer,
>
> Thank you for your valuable feedback and for recognizing the merits of our dataset's design. Should our subsequent responses not fully address your concerns, we remain open to ongoing dialogue with you.
>
> ### (1) Details of the Manual Review:
> We concur that a more thorough exposition of the manual review process is paramount for a clearer understanding. In our initial submission, due to space limitations, we couldn't delve deeply into this area. However, upon receiving your suggestions, we have expanded upon this segment.
>
> To elaborate:
>
> We collect 12,331 GUI pairs in the automated collection. Due to the inherent constraints of ADB, contemporary data collection mechanisms frequently falter in retrieving metadata for the UI type WebView as well as certain bespoke third-party UI components. Concurrently, for certain obscured Android fragments and UIs, only the metadata for the foreground UI is of relevance, given that the overlaid UI remains undisplayed on the prevailing screen. Yet, extant data collection tools might inadvertently accumulate data from both the visible and concealed UI, leading to potential data inaccuracies.
>
> To address these challenges, an additional round of manual data validation is undertaken by a panel of three specialists, each boasting a minimum of one year's expertise in Android development. These individuals meticulously scrutinize each pair against two pivotal metrics: data integrity and pair coherence.
> This manual validation encompass a meticulous verification of harvested pairs, identification and elimination of pairs with erroneous metadata, and an assessment of the logical consistency of the matched pairs.
> Throughout this rigorous verification phase, the evaluators also actively engage in the manual matching of certain phone-tablet GUI pairs.
> In this round of manual verification, we remove 2,296 pairs with obvious visual mismatches between screenshots and their metadata.
>
> ### (2) Figure 7's Modification:
> We greatly appreciate your feedback on Figure 7. We concede that its current representation could lead to potential confusion. Acting upon your recommendation, we have now segregated the two phone GUI-generated tablet GUI examples into separate subfigures, enhancing clarity and ease of interpretation.
>
> ### (3) Results Analysis and Major Research Questions:
> Additionally, to shed more light on the challenges and complexities of GUI conversion, we've incorporated in Section 5.6 [Research Questions and Observations from Experiments] of the original paper. This section elucidates the observed challenges, offers in-depth analyses of the outcomes, and highlights potential avenues of research that could augment the effectiveness of this task.
>
> To elaborate:
> As illustrated in Table 2, even the best-performing model, VTN, manages to accurately reproduce a mere 20\% of the UI components present in the ground truth of the corresponding table.
> Notably, the left example in Figure 7 depicts a scenario where our proposed method exhibits a relatively superior GUI generation.
> However, the instances on the right are notably subpar. We hypothesize that this disparity arises due to the intricate containment and nesting relationships among UI components in the right example, resulting in a more convoluted metadata structure in comparison to the left. Consequently, discerning the relationships between UI components proves challenging for the model.
> From our preliminary experiments and insights, we highlight key research queries that currently permeate the domain, aiming to advance the field:
>
> **GUI Encoding and Representation**: Our paper delves into the intricate nature of GUI, a complex data structure encompassing a wide variety of structural information and metadata. The methods of representing and encoding these GUI data are instrumental in facilitating tasks such as data dimensionality reduction and multimodal data fusion, potentially leading to a significant enhancement in the generation effect. This complex encoding presents a ripe area for further exploration and potential innovation.
>
> **Modeling UI relationship**: Furthermore, the logical relationship between UI components, such as those images arranged side by side or images with texts below, provides an exciting avenue for research. Defining and modeling these relationships could pave the way for a new understanding of the interplay between UI components and could have broader applications in automated GUI development.
>
> **Addressing Non-correspondence Contents**: The fragmentation of Android screens poses challenges, particularly when a tablet GUI in a phone-tablet pair houses more content than its phone counterpart.
> Navigating these content discrepancies presents a valuable research trajectory, with implications for enhancing cross-device GUI adaptability.
>
> Thank you once more for your invaluable insights, and we believe these refinements amplify the paper's clarity and depth.

---

### Author Response · Authors · 2023-08-13
**General Response to All Reviewers**

Dear Reviewers,

We deeply appreciate the time and effort you have dedicated to reviewing our manuscript. Your comprehensive feedback has been instrumental, underscoring both our work's strengths and areas where refinement was needed. Your acknowledgment of our contribution is truly encouraging, and we are keen to address the concerns you've highlighted.

Below, we outline the primary amendments made during the rebuttal process. For your convenience, changes in the revised manuscript are indicated using the blue font.

### (1) Pairwise Data Collection Approaches:
We are grateful for the invaluable input on our methods section. In response, we've relocated the methods section from the supplementary material to the main body of the paper. To accommodate this, and in line with your recommendations, adjustments were made to the content for a seamless fit. Further, to emphasize our paper's significance, we've refined the title from ‘Papt: a pairwise GUI Dataset between Android Phones and Tablets’ to ‘Pairwise GUI Dataset Construction for Android Phones and Tablets’.

### (2) Enhanced Experimental Analysis and Major Research Questions:
We acknowledge, and as stated directly within the manuscript, that our current model requires further enhancement. This is, in fact, one of the primary motivations for our selection of the dataset track. In response to the feedback, we have conducted more analysis of the existing model’s results, which we have included in Section 5.6 of the revised manuscript. Furthermore, we have discussed the major research questions related to GUI automated development, informed by our current undertakings and experiences.
One of our primary objectives remains to release our datasets and tools to the public domain. We hope this initiative will inspire and facilitate further research in this area, representing the paper’s contribution to the scholarly community.


### (3) Significance of the Dataset for Research:
The introduction has been enhanced to underscore the profound significance of our dataset for ongoing research. In the supplementary material, a dedicated section elaborates on the potential contributions of our dataset to existing research endeavors.

### (4) Ethical Considerations:
In our commitment to ethical research, we've delineated potential ethical concerns associated with our dataset in the supplementary section.

We have further made several modifications in the paper in direct response to individual reviews, and these changes are detailed in the corresponding responses to those reviews.

Once again, thank you for your expertise and insights. Your feedback has undoubtedly enriched our work.

---

### Decision · Program_Chairs · 2023-09-22

**Decision:**

Accept (Poster)

**Comment:**

All reviewers are excited about this dataset, while have concerns about the size of the
data set and that it only focuses on Android applications. Balancing pros and cons, we
recommend accept of this work, while encouraging the authors to continue enrich both
the size and the diversity of this dataset.